# A Novel Order Analysis and Stacked Sparse Auto-Encoder Feature Learning Method for Milling Tool Wear Condition Monitoring

**DOI:** 10.3390/s20102878

**Published:** 2020-05-19

**Authors:** Jiayu Ou, Hongkun Li, Gangjin Huang, Qiang Zhou

**Affiliations:** School of Mechanical Engineering, Dalian University of Technology, Dalian 116024, China; oujy@mail.dlut.edu.cn (J.O.); huanggj@mail.dlut.edu.cn (G.H.); zhouqdut@mail.dlut.edu.cn (Q.Z.)

**Keywords:** order analysis, stacked sparse autoencoder, spindle current signals, tool wear condition monitoring

## Abstract

Milling is a main processing mode of the modern manufacturing industry, which seriously affects the quality and precision of the machined workpiece. However, it is difficult to monitor the tool wear condition in the continuous cutting process, especially under a variable speed condition. The existing tool wear condition monitoring methods only carry out analysis with a constant engine speed. Different from the general monitoring methods, this paper put forward a milling cutter wear condition monitoring method based on order analysis (OA) and stacked sparse autoencoder (SSAE). The methodology in the research include signals feature extraction and tool wear state monitoring and were designed to analyze the three-phase spindle current signals instead of the traditional force signals and vibration signals. The variable speed signals were transformed into angle domain stationary signals by order analysis, and the SSAE neural network was used to monitor the tool wear state. The proposed method was verified on the laboratory signals and the results showed a better performance than the other methods and a better applicability in actual industrial manufacturing.

## 1. Introduction

Milling cutter wear is an inevitable tool degradation phenomenon caused by mechanical, thermal, chemical and abrasive particles acting on the workpiece, which will lead to the decline of product quality and the increase of production cost [1]. The mechanical system is often disturbed by the outside world [2]. As time goes on, wear and damage are inevitable [3]. Statistics show that 20% of machine stops are due to tool wear [4,5]. Using tool condition monitoring technology can increase productivity by ~10%–60% and equipment utilization by more than 50% [6]. Therefore, monitoring of the state of tool wear is a pressing need in actual production, which can minimize downtime, workpiece damage and prevent catastrophic injury.

Traditional tool wear monitoring methods are usually time-frequency analysis and multi-sensor fusion [7,8]. Among these various methods, vibration signals, cutting force signals and acoustic emission signals are widely used. The relationship between tool wear and signal characteristics is established by sophisticated mathematical interpretation. Zhu et al. [9] developed an original tool-monitoring method named morphological component analysis (MCA) model based on the full tool wear surface area. In the case of serious noise, fuzzy boundary and tool wear image dislocation, the region-growing algorithm is used to detect the defect and extract the wear region from the target tool image. The results show that MCA algorithm can effectively extract the wear area of the target tool image, and the estimation of the wear area can extend the estimation method of tool wear width, which is more accurate than the other methods. Zhang et al. [10] used multiple sensor data, such as cutting vibration data and power data, as well as actual processing parameters to develop a system to realize tool condition-monitoring and life estimation efficiently, which is widely used in small and medium-sized enterprises. Uekita et al. [11] presented a tool condition monitoring technique combining short-time Fourier transform and spectral kurtosis analysis to identify chatter process, which will guarantee the machining precision of large-size components and provide tool state information. Javier et al. [12] proposed a Gaussian mixture model based on dynamic probabilistic clustering for tool condition monitoring. It demonstrates that the dynamic component selection feature can detect the optimal number without relying on the initial estimate. These methods still mainly depend on professional experience and knowledge. With the increasing demand of industry precision, the methods will become more and more complex and less adaptive, which prevents the signal information being fully extracted and is only suitable for some particular machining conditions. The traditional methods are not suitable for the tool wear monitoring of large rotating machinery. Thus, how to extract more effective information from the signals to reflect the degree of tool wear must be taken into consideration.

In recent years, neural network models such as the convolutional neural network (CNN), artificial neural network (ANN) and the sparse autoencoder (SAE) have developed rapidly in the fault diagnosis of rotating machinery due to its incomparable advantages and made it possible to process data in large quantities [13,14]. Cao et al. [15] presented an intelligent tool wear state recognition technique combining derived wavelet frames (DWFs) with a convolutional neural network (CNN) using vibration signals. DWFs are used to decompose the original signals into frequency bands and different center frequencies. The CNN model is applied to train and test two-dimensional signals. This method reduces the need for expert knowledge and improves recognition accuracy, generalization and the calculation efficiency. Cheng et al. [16] used the deep convolutional neural network (DCNN) to monitor the abrasive belt wear based on the acoustic signals. It overcame the difficulties of tool condition monitoring due to the unknown number of abrasive particles, random shape changes and complex sound signals. The one-level precision reached 97.6%. Patra et al. [17] discussed the tool wear behavior of micro-drills in the process of micro-drilling. A neural network model was established to predict the number of holes with thrust and cutting conditions as the input parameters, and was applied to the new cutting conditions to verify the performance of the neural network model in tool state prediction. Ochoa et al. [18] diagnosed the cutter wear condition through the stacked sparse autoencoder (SSAE) neural network with acoustic emission signals in high-speed machining. Through discussing the performance of the different signals obtained from different sensors, it showed that the accuracy of the acoustic signals in the monitoring of the tool wear state had a high diagnostic accuracy of 99.6%. It can be found from the existing literatures that most of the methods directly input the time-domain signals or frequency-domain signals into the deep learning model as sample data. However, these methods are mainly aimed at the analysis of stationary signals with a constant engine speed. Less research has been conducted on the recognition of tool wear state based on variable speed signals. In the process of impeller machining, the current, force and vibration signals are often non-stationary signals and frequency aliasing easily occurs. Therefore it is an urgent need to develop a method for tool wear state monitoring under variable speed conditions.

In this paper, a novel feature learning method called order analysis and stack sparse autoencoder (OA-SSAE) was proposed for tool wear condition monitoring. It was applied to analyze the spindle current signals of computerized numerical controlled (CNC) machine tools and can be summarized as two flowing parts: the preprocessing of variable speed signals and tool wear state monitoring. In the first part, order analysis is used to resample the variable speed signals into angle domain stable signals. The order spectrum is obtained through the Fourier transform and the order features are extracted as samples. In the second part, the stacked sparse autoencoder is adopted to predict and classify the tool wear states. The effectiveness was verified on the laboratory signals and the results showed that the proposed method had a better classification effect than the other methods.

The rest of this paper is organized as follows: Section 2 presents a detailed overview of the tool wear state-monitoring OA-SSAE model combining order analysis (OA) and SSAE. Section 3 introduces the research idea and algorithm flow of this paper. Section 4 describes the results of the proposed method verified on laboratory signals. The conclusions and expectations are given in Section 5.

## 2. Basic Theory

### 2.1. Order Analysis

In the process of impeller machining, different spindle speeds are needed to cooperate with cutting. When the speed changes significantly, if the collected time-domain signals are directly transformed into the frequency-domain characteristics through the Fast Fourier Transform (FFT) analysis, the different frequency features are mixed together, then the frequency aliasing effect will occur in the spectrum, which makes the frequency domain information impossible to distinguish. There are therefore some defects in using the traditional spectrum characteristics to monitor the wear state of a milling cutter.

Potter adopted order analysis to solve the problem of variable speed signals in 1989 [19]. It is clear that the law of different order components changed with speed and removed the influence of speed change on the spectrum distribution [20]. The main principle of order analysis is the angle domain sampling theorem, which mainly consists with two sampling processes [21]. The first process is equal time interval sampling. The current signals and spindle rotational speed pulse signals need to be sampled with a constant sampling frequency at equal time intervals. The pulse signals of the spindle rotational speed is collected by the key-phasor sensor. The second process is interpolation and resampling. According to the speed pulse signals, calculating the time series of the equal angle sampling of speed signals, the original current signals are sampled at equal angle intervals. The time domain non-stationary signals are transformed into the stationary signals in angle domain [22]. The corresponding frequency spectrum is obtained through the Fourier transform, which is the order spectrum. Angular sampling needs to meet the sampling theorem:(1)$$\left\{ \begin{array}{l} \Delta o=\frac{1}{r}=\frac{1}{N_{\partial }\times \Delta \theta }\\ o_{c}\leq \frac{o_{s}}{2},o_{s}=\frac{1}{\Delta \theta } \end{array}\right.$$
where Δo is order resolution, r is the sampling rotation speed of the main shaft, $N_{\partial }$ is the corner sampling point number, Δθ is the corner-sampling interval, $o_{c}$ is the signal analysis order and $o_{s}$ is the angular sampling frequency.

In order to determine the resampling time, it is generally assumed that the central axis rotates in a small period of time. According to this assumption, the spindle rotation angle θ can be written as a polynomial about time t [16]:(2)$$\theta \left(t\right)=b_{0}+b_{1}t+b_{2}t^{2}$$
where $b_{0},b_{1},b_{2}$ are the coefficients of polynomials, which can be calculated by three consecutive key-phase signals $t_{1},t_{2},t_{3}$. Supposing that the increment of the spindle angle is a constant number Δφ in the adjacent key-phase, then:(3)$$\left\{ \begin{array}{l} \theta \left(t_{1}\right)=0\\ \theta \left(t_{2}\right)=\Delta \varphi \\ \theta \left(t_{3}\right)=2\Delta \varphi \end{array}\right.$$
(4)$$\left(\begin{array}{c} 0\\ \Delta \varphi \\ 2\Delta \varphi \end{array}\right)=\left(\begin{array}{ccc} 1 & t_{1} & t_{1}^{2}\\ 1 & t_{2} & t_{2}^{2}\\ 1 & t_{3} & t_{3}^{2} \end{array}\right)\left(\begin{array}{c} b_{0}\\ b_{1}\\ b_{2} \end{array}\right)$$

Through the above two formulas, the time t corresponding to any corner in $\left[\begin{array}{cc} 0 & 2\Delta \varphi \end{array}\right]$ can be obtained, as shown in the following formula:(5)$$t=\frac{1}{2b_{2}}\left[\sqrt{4b_{2}\left(\theta -b_{0}\right)+b_{1}^{2}}-b_{1}\right]$$

According to equal angle sampling, if the sampling interval is Δθ, then the number of sampling points is an integer and the corresponding sampling angles can be expressed as follows: 0,Δθ,2Δθ⋯kΔθ.

### 2.2. Stacked Sparse Autoencoder

Autoencoder (AE) is an unsupervised deep learning network [23]. It designed to make the reconstruction errors minimal and use the low-dimension features to replace the high-dimension input signals [24,25]. Autoencoder is a single hidden layer neural network, and the schematic diagram is presented in Figure 1.

For an unlabeled dimensional training sample $X=\left[X_{1,}X_{2}\cdots X_{n}\right]\in R^{m\times n}$. m and n can be any value, and any samples can be expressed as $X_{i}={\left[x_{1},x_{2},\cdots x_{m}\right]}^{T}\in R^{{}^{m\times 1}}$.

The first layer aims to connect the input layer and the hidden layer and obtain a hidden feature vector $h={\left[h_{1},h_{2},\ldots ,h_{d}\right]}^{T}\in {ℜ}^{d\times 1}$ using an activation function [26]. The most commonly used activation function is the sigmoid function, which is expressed as
(6)$$sig(t)=\frac{1}{1+e^{-t}}$$
(7)$$h=sig\left(W_{1}\cdot x+b_{1}\right)$$
where $W_{1}$ is a weight matrix, $b_{1}$ is bias vector and $\theta_{1}=\left\{W_{1},b_{1}\right\}$ means a set of the first layer parameters.

Then, the second layer transforms the hidden feature vector into an output vector $\tilde{X}=\left\{{\tilde{X}}_{1},{\tilde{X}}_{2},{\tilde{X}}_{3},\ldots ,{\tilde{X}}_{h}\right\}\in {ℜ}^{m\times n}$,${\tilde{X}}_{i}={\left[{\tilde{x}}_{1},{\tilde{x}}_{2},\ldots {\tilde{x}}_{m}\right]}^{T}\in {ℜ}^{m\times 1}$ similar to the process of the first layer, which can be expressed as
(8)$$\tilde{x}=sig\left(W_{2}\cdot y+b_{2}\right)$$
where $\theta_{2}=\left\{W_{2},b_{2}\right\}$ is the parameter set of the second layer.

The output of the second layer is a less dimensional eigenvector, which contains the information of the original input signals and can be reconstructed back to the original signals. The purpose of the autoencoder network training is to select a suitable parameter set $\theta =\left\{\theta_{1},\theta_{2}\right\}=\left\{W_{1},b_{1};W_{2},b_{2}\right\}$ to make the reconstruction error minimal [27]. Generally, the mean square error cost is used to represent the size of the reconstruction error. The reconstruction error can be expressed as
(9)$$E\left(\theta \right)=\frac{1}{m}\sum_{i=1}^{m}\left(\frac{1}{2}{\left\|{\tilde{x}}_{i}-x_{i}\right\|}^{2}\right)$$

Meanwhile, the sparsity constraint is added to avoid overfitting when the training data dimension is massive [28]. The sparse penalty term works as a switch, and if its output value is nearer to 1 then the neuron is considered to be active, otherwise it is considered to be inactive if the value is closer to 0 [29]. Then, the average activation could be expressed as
(10)$${\hat{\rho }}_{j}=\frac{1}{m}\sum_{i=1}^{m}\left[a_{j}^{\left(2\right)}X_{i}\right]$$

In the hidden layer, in order to meet the sparsity limitation, the average activation degree $\hat{\rho }$ of the hidden layer neurons is taken as a value ρ closely to zero, meaning that most neurons are inactive. Then, the penalty term is:(11)$$J_{sparse}=\beta \sum_{j=1}^{s_{2}}KL\left(\rho \|{\hat{\rho }}_{j}\right)$$
where β is the weight adjustment parameter and $s_{2}$ is the neurons number of the second layer.

There are many forms of sparse restriction. Usually $KL\left(\rho \|{\hat{\rho }}_{j}\right)$ divergence is chosen to represent the penalty factor of the hidden layer neuron, which can be written as
(12)$$KL\left(\rho \|{\hat{\rho }}_{j}\right)=\rho \log\frac{\rho }{{\hat{\rho }}_{j}}+\left(1-\rho \right)\log\frac{1-\rho }{1-{\hat{\rho }}_{j}}$$

As the difference between ρ and $\hat{\rho }$ increases, the value of the penalty factor will rise sharply [30].

Furthermore, to avoid overfitting, a weight decay term $J_{Weight}$ is applied:(13)$$J_{Weight}=\frac{\lambda }{2}\sum_{l=1}^{2}\sum_{i=1}^{s_{l}}\sum_{i=1}^{s_{l+1}}{\left(W_{ji}^{\left(l\right)}\right)}_{i}^{2}$$
where λ is the weight adjustment parameter and $s_{l}$ is the unit number of layer l.

In summary, the overall cost function of the system can be expressed as
(14)$$L\left(\theta \right)=E\left(\theta \right)+J_{sparse}\left(\theta \right)+J_{weight}\left(\theta \right)$$

The loss function consists of three parts: reconstruction error, weight decay term and sparse restriction term [31,32]. A single SSAE network can only carry out feature learning with no classification function. It can only learn the features of different tool wear states, but it cannot classify them. It therefore needs to add a classifier to the last layer of the SSAE network to complete the classification of tool wear states. The stacked sparse autoencoder with two hidden layers and a SoftMax classifier is shown in Figure 2.

Through repeated experiments, the two hidden layer neural network was selected in this paper. If the single-layer neural network was used, the learning ability of the network would be insufficient, and the effective information could not be completely extracted from the signals, which greatly reduces the diagnostic ability of the model and which could not accurately determine the tool wear state. If a neural network with three or more layers was selected, then the accuracy would not be significantly different from the two layer neural network, but the computed time of the network would obviously increase. A two hidden layer neural network can ensure a high accuracy and minimal computing time.

## 3. General Procedure

In this paper, a novel method named order analysis and stacked sparse autoencoder (OA-SSAE) was developed for milling tool wear condition monitoring. The optimization algorithm of the network parameters was Large-Broyden, Fletcher, Goldforb, Shanno (L-BFGS), which has the advantages of fast convergence and low memory cost and is widely used in the parameter selection of neural networks. The flow diagram of the main idea of this paper is presented in Figure 3, and the general algorithm implementation steps are described as follows:

**Step1**. Order analysis is performed on the original current signals, and extract the order characteristics, which contain almost all the information of the original signals.

**Step2**. The order characteristics are used as the input of the SSAE network and the network key parameters are optimized by L-BFGS iteration, then the eigenvector of the first hidden layer is obtained.

**Step3**. The feature vector obtained in the previous step is used as the input of the second layer of the SSAE network, and the same method is used to train the network parameters and the output of second hidden layer is obtained.

**Step4**. The parameters of the SoftMax classifier are initialized and the output vectors obtained in Step3 are used as input samples. Then the labels of the tool wear data are combined to train the SoftMax classifier and optimize a suitable set of parameters by L-BFGS iterative optimization.

**Step5**. The SSAE and SoftMax are taken as a whole, then the overall cost function and the overall partial derivative function are calculated for each parameter.

**Step6**. The above metrics are adopted as the initialization parameter values of the whole network, and then the L-BFGS algorithm is used to iteratively find a parameter value closest to the above cost function, which is the optimal parameter value of the overall model.

**Step7**. The testing sample data are input into the whole model to classify tool wear so as to judge the wear type and analyze the results.

## 4. Experimental Verification

For certificating the superiority and feasibility of the developed method in this paper for tool wear state monitoring, an experiment test was performed on a three-axis CNC machine tool to collect the tool wear full-life spindle current signals, and the wear states of the tool were measured at different times.

### 4.1. Tool Wear State Test Rig

The machine tool used in this experiment was a CMV-850A vertical machining center with a Fanuc 0I-MC numerical control system produced by the Taiwan Dongyu Precision Machine Company. The spindle motor was a beta 12/8000i three-phase four-pole asynchronous motor produced by the Fanuc Company. The spindle drive mode was a synchronous belt drive with a transmission ratio of 1. The sensor used in the experiment was the LT 108-S7 closed-loop Hall current sensor produced by the Swiss LEM Company. The sensor installation is presented at Figure 4, and the main performance parameters of the sensor are shown in Table 1

The tool was a Taiwan Di brand two-edged cemented carbide flat end milling cutter. Its diameter was 12 mm and its helix angle was 30 degrees. The workpiece material was 40Cr and the dimensions were 90 mm × 120 mm × 70 mm, as can be seen in Figure 5.

This experiment used dry cutting without cutting fluid and the sampling frequency was 4096 Hz. The collected signals made a total of 1040 samples. According to the wear amounts on the back face of the cutting tool, the original current signals were divided into five different wear states. Table 2 shows the numbers in each of the five different wear states samples. Moreover, one second domain signals of five different kinds of tool wear states are shown in Figure 6.

### 4.2. Result and Analysis

Considering the variable speeds condition in the impeller processing, this paper analyzed the order of each sample and obtained the order spectrum through the FFT, as shown in Figure 7. Through the analysis of the order spectrum, it could be found that the amplitude of the first 25 orders were more obvious, which contained most of the information of the signals, while the component amplitude greater than 25 orders was almost 0, which contained little information of the signals and were not worth considering. Therefore, in this paper, 330 features of the first 25 orders were selected and made 330 × 1024 samples as the input to train the neural network.

For proving the feature extraction ability of the order analysis, the collected data were inputted into the SSAE-SoftMax model in two ways: one was to make 4096 × 1024 samples of the collected time domain original signals and input them directly into the SSAE network; the other was to analyze the order of each sample signal and obtain the order characteristics as the input of the SSAE network. The comparison results of the two methods in different tool wear state recognition, which is shown in Figure 8. From this, it can be observed that the accuracy of the neural network recognition with order features as the sample was higher than that of original current signals. The recognition accuracy of the neural network with the order features as the sample for the five wear states of the tool was over 95%.

To further prove the effectiveness of the order analysis method for the feature extraction of the variable speed current signals, principal component analysis (PCA) was applied on the three-dimensional visualization of the different tool wear states shown in Figure 9. As can be drawn from Figure 9a, if the collected original data is directly input into the SSAE neural network without any processing, the clustering analysis is not effective, except that the state 5 (severe wear) can be separated, and the other four states are mixed together. However, the data processed by order analysis can be better separated into five different states, as shown in Figure 9b, but it still can be found that the separation effect of state 1 and state 2 are not as good as the other three states. This is because there was no clear distinction between the cutting state of the new cutter and the initial wear state of the cutter itself, so the difference of the data characteristics were not obvious. Comparing the two cluster analysis visual effect maps, it was clear that the data classification effect processed by the order analysis was better than that of the original data.

To investigate the performance of the proposed method in different tool wear states, the multi-class confusion matrix is shown in Figure 10. The confusion matrix is the most intuitive method to measure the accuracy of a classification model. The multi-class confusion matrix is used to count the number of observation values of the classification model, which is classified into a wrong category and a right category, respectively, and then the results are displayed in a table. The ordinate axis of the confusion matrix represents the actual label of classification, and the horizontal axis represents the predicted label. Therefore, the element on the main diagonal of the confusion matrix indicates the classification accuracy of each condition. From Figure 10, we can find that among all kinds of states, state 1 (initial wear), state 2 (intermediate wear I) and state 5 (severe wear) were the best recognition ones with a recognition accuracy over 0.99.

Besides, five similar methods were used to handle the same spindle current signals for comparison, namely the extreme learning machine (ELM), back propagation neural network (BPnn), support vector machine (SVM), radio frequency (RF) and the k-nearest neighbor (KNN) algorithms. Ten identical trials are performed for recognizing the tool wear state samples. The architecture of the proposed method was 330-100-120-5 by repeated trials. Here, 330 is the number of neurons of the input layer, 100 is the number of neurons of the first hidden layer, 120 is the number of neurons of the second hidden layer and 5 is the number of output layer. The outcomes of the six methods are shown in Figure 11. The detailed results are listed in Table 3. The input of all the comparison methods are the same order analysis data, and the computation time is the time of wear classification stage.

From Figure 11, in the ten repeated experiments, the average accuracy of the proposed method was higher than that of other methods. The recognition accuracy of the traditional BPnn and ELM algorithm was small, only about 80%. Although the average accuracy of the SVM and the RF algorithm reached more than 90%, the average accuracy was not stable. For example, the accuracy of SVM in the sixth experiment was higher, while the accuracy of the third experiment was slightly lower. Therefore, compared with the other methods, the accuracy of the proposed method was higher and the model was more stable. It can be concluded that the proposed model shows a superiority in tool wear state recognition.

From Table 3, we can see that the average training accuracy of the proposed method was 96.41%, higher than the contrasted five methods, which were 73.27%, 80.38%, 89.87%, 90.24% and 84.14% respectively. The average testing accuracy of the proposed method was 98.79%, which was much higher than that of the other four methods, which were 79.18%, 81.65%, 91.83%, 94.36% and 87.18%, respectively. As for the computation time, the proposed method was 16.934 s, which was higher than the other methods, which obtained 6.189 s, 1.180 s, 8.329 s, 15.436 s and 5.317 s, respectively. Although the proposed method took longer than the other methods, the recognition accuracy was the highest. This means that the proposed method shows better classification ability for tool wear states.

In this paper, the SSAE neural network was applied for tool wear state monitoring. In order to explore the effect of the number of neurons of the second hidden layer’s on the recognition accuracy and computation time, ten experiments were carried out, in which the number of hidden layers gradually increased evenly. The impact of the number of neurons of the second hidden layer of the SSAE on the recognition accuracy and computation time is shown in Figure 12. It can be found from Figure 12, with the increase of the number of neurons in the hidden layer, the average testing accuracy and computation time increases accordingly. When the number of neurons increases to 600, the average testing accuracy reaches the highest level. After that, even if it rises, the testing accuracy does not change significantly, but the corresponding calculation time obviously increases. Therefore, the most suitable number of neurons in the second hidden layer was 600, which could obtain the highest testing accuracy and the least calculation time.

## 5. Conclusions

This paper analyzed a data preprocessing method to improve the recognition effect of the deep learning diagnostic model, and put forward a method of extracting features by order analysis from the collected original signals, which reduced the model computational complexity and improved the efficiency of monitoring and classification. From the experiment and comparison results, the proposed SSAE neural network combined with the order analysis model can better extract the in-depth features of tool wear. Followed by the SoftMax classifier, which can better realize the classification and recognition of tool wear states, it provides more accurate reference data for the online monitoring of milling tool wear states.

It is certain that the proposed method will be further studied and applied to more complex operating conditions in the process of variable speed milling. Furthermore, future research will focus on the optimization mechanisms of the model parameters and the study of the multi-parameter fusion of variable speed.

## Figures and Tables

**Figure 1 sensors-20-02878-f001:**
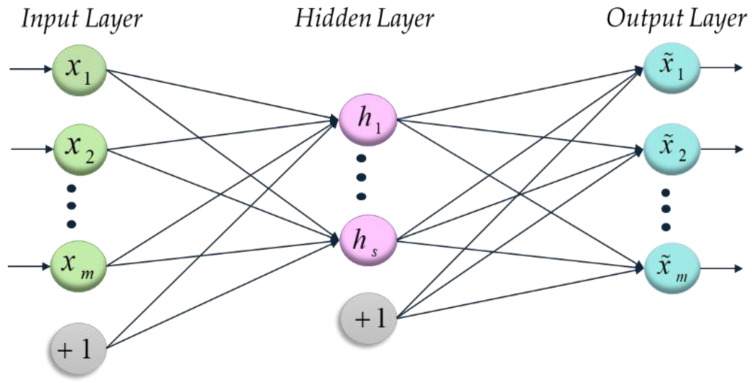
The structure of a standard autoencoder with a single hidden layer.

**Figure 2 sensors-20-02878-f002:**
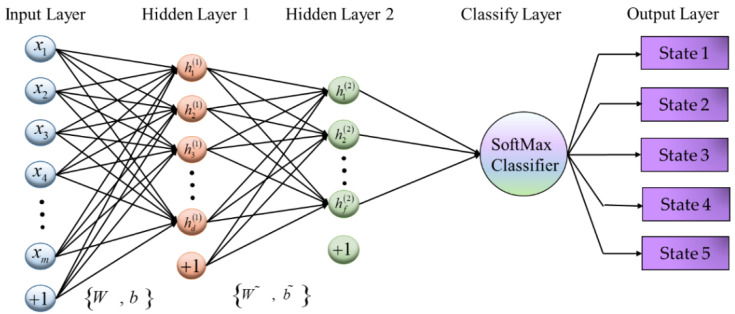
The structure of the two hidden layer sparse autoencoder with a SoftMax classifier.

**Figure 3 sensors-20-02878-f003:**
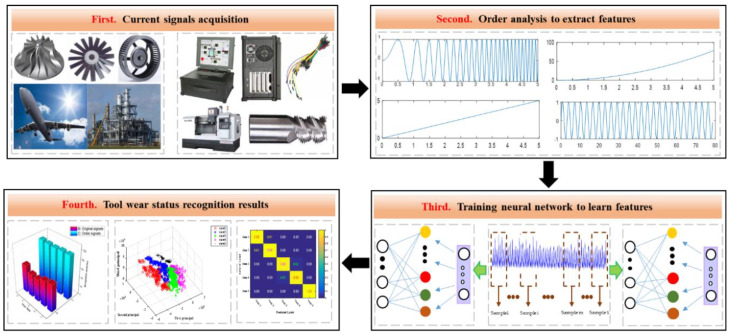
The structure of the proposed method.

**Figure 4 sensors-20-02878-f004:**
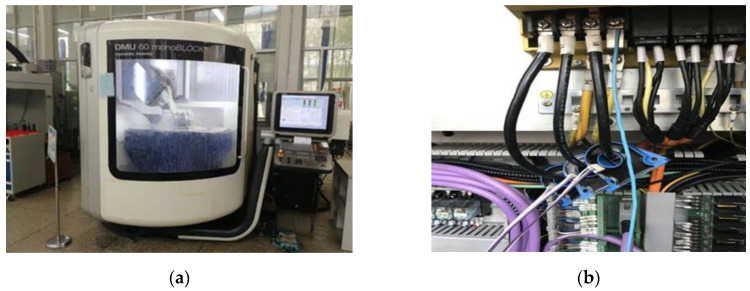
Experiment computerized numerical controlled (CNC) milling machine and the current sensor installation procedure: (**a**) the three-axis CNC milling machine; and (**b**) the Hall affect current sensor installation.

**Figure 5 sensors-20-02878-f005:**
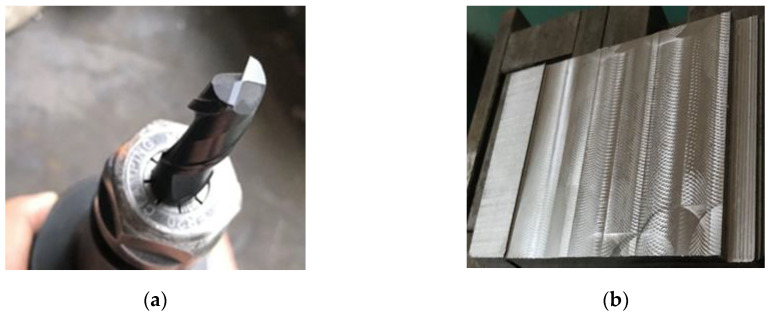
The milling tools and workpieces. (**a**) The two-edged flat end milling cutter. (**b**) The workpiece in the experiment.

**Figure 6 sensors-20-02878-f006:**
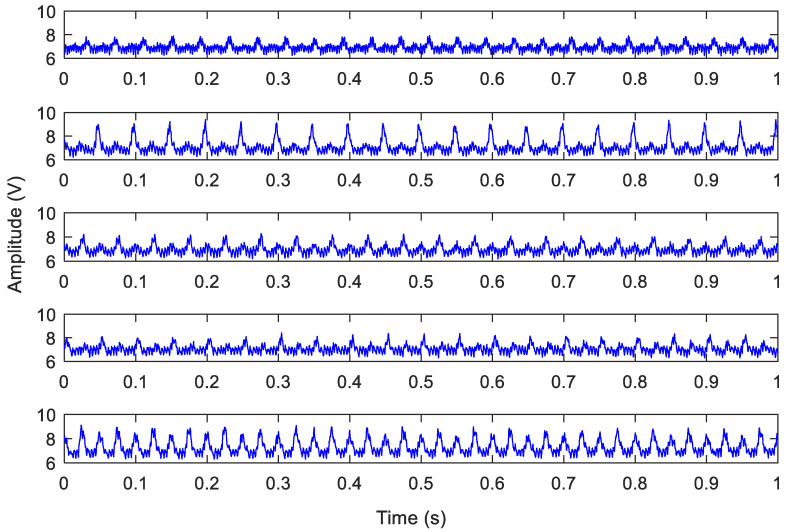
Current signals of the five different tool wear states.

**Figure 7 sensors-20-02878-f007:**
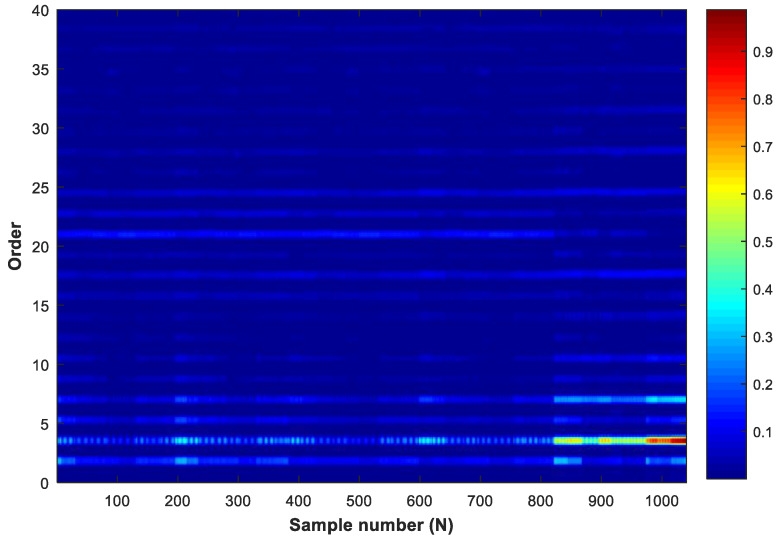
The picture of the order spectrum.

**Figure 8 sensors-20-02878-f008:**
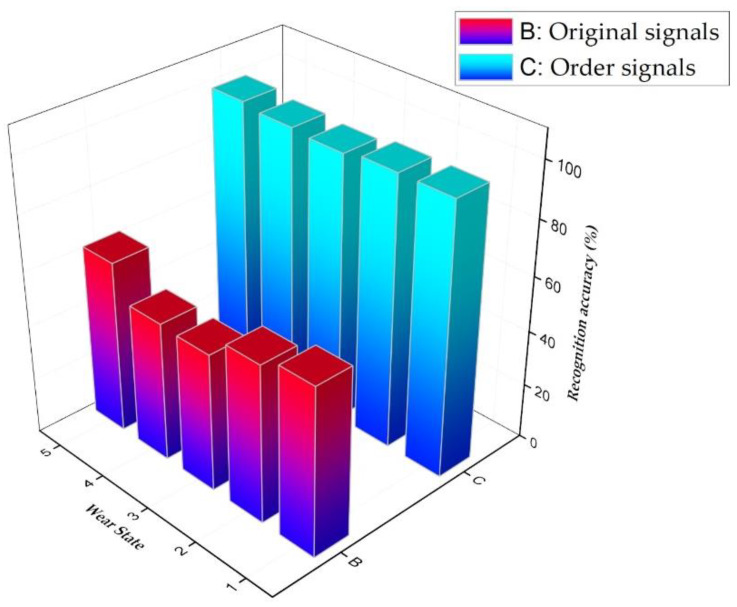
The results of the comparison of the two methods.

**Figure 9 sensors-20-02878-f009:**
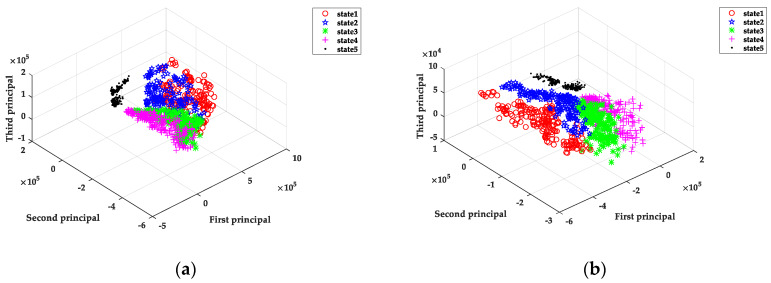
Three-dimensional visualization for the different tool wear states. (**a**) The original data principal component analysis visualizing picture. (**b**) The data processed by order analysis visualizing picture.

**Figure 10 sensors-20-02878-f010:**
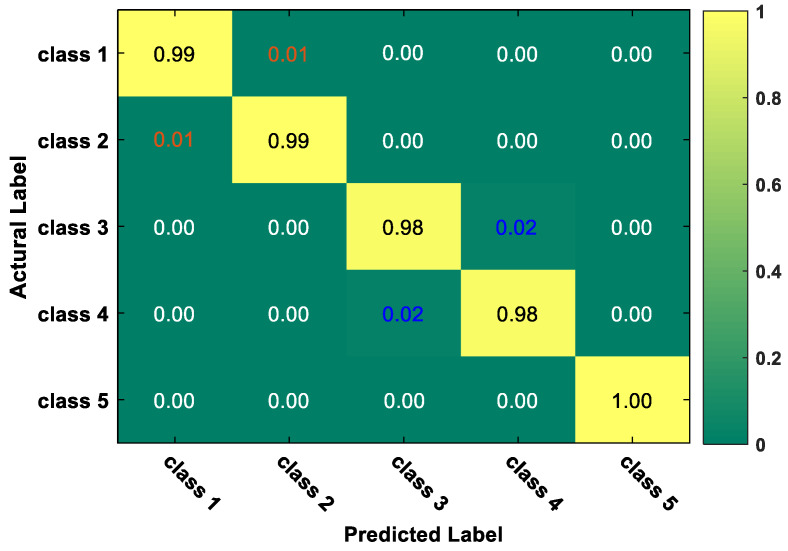
The multi-class confusion matrix of the proposed method.

**Figure 11 sensors-20-02878-f011:**
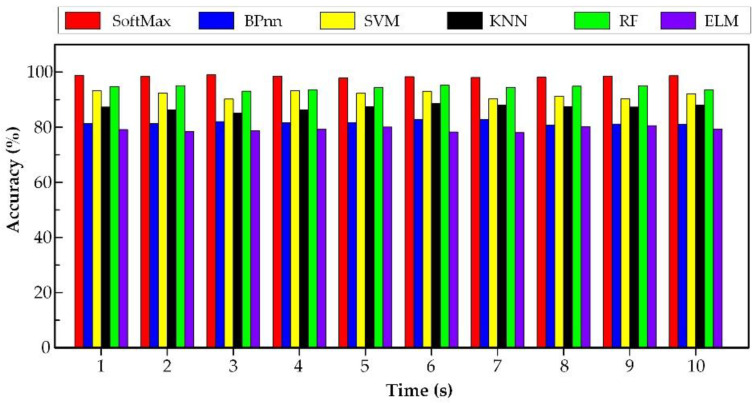
The accuracy of the six methods.

**Figure 12 sensors-20-02878-f012:**
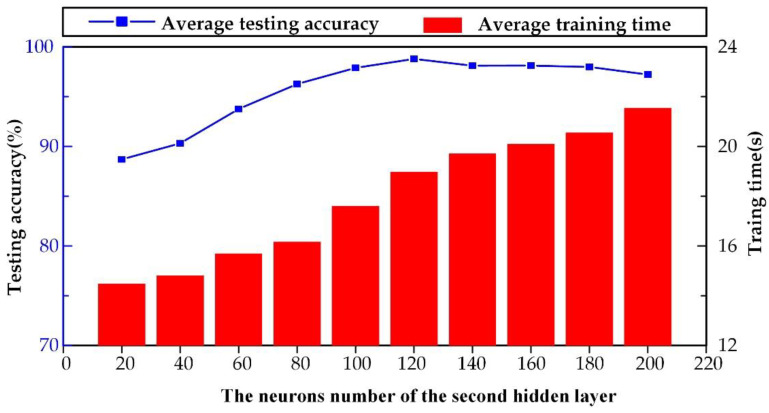
The results of the effect of number of neurons of the second hidden layer on the recognition accuracy and calculation time.

**Table 1 sensors-20-02878-t001:** The main performance parameters of the closed-loop Hall current sensor.

Item	Parameter
Induced current type	AC or DC or Pulse current
Band width	100 kHz
Response time	<1
Nominal output	50 mA
Di/dt tracing accurate	Better than 100 A/us
linearity	<0.1%

**Table 2 sensors-20-02878-t002:** Description of the five different tool wear state samples.

Tool Wear State Sample	Initial Wear	Intermediate Wear I	Intermediate Wear II	Intermediate Wear IIII	Severe Wear
Training	157	147	176	183	177
Testing	40	40	40	40	40
Labels	1	2	3	4	5

**Table 3 sensors-20-02878-t003:** Description of the compression average accuracy.

Method	Training Accuracy (%)	Testing Accuracy (%)	Computation Time(s)
SSAE-Softmax	96.411	98.788	16.934
ELM	73.268	79.176	6.189
BPnn	80.379	81.654	1.180
SVM	89.873	91.831	8.329
RF	90.421	94.364	15.436
KNN	84.142	84.182	5.317

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
