# Peer review of "A Novel Order Analysis and Stacked Sparse Auto-Encoder Feature Learning Method for Milling Tool Wear Condition Monitoring"

_sensors, 2020, doi:10.3390/s20102878_

Round 1
Reviewer 1 Report
The abstract is not abstract in its essence. Authors are asked to skip the methodology’s description in the abstract and clarify what the contribution consists of.
Spelling mistakes occur several times.
Fig. 3., 6. have to be enlarged.
How was the fig. 10 obtained? It should be described as well.
The literature review must be developed.
It is not clear what the sentence or actually ”330-100-120-5” means: ”The architecture of SSAE-SoftMax method is 330-100-120-5 by repeated trials.”
Results given in fig. 11. must be described in details, apart from percentages and statistics.
Section 6 title suggest a patent, however it is empty.
For the milling machine Authors might include availability aspects as it is mentioned in Kostrzewski, M.; Gnap, J.; Varjan, P.; Likos, M. Application of simulation methods for study on availability of one-aisle machine order picking process. Communications. 2020, 22, 2, pp. 107-114. https://doi.org/10.26552/com.C.2020.2.107-114
Author Response
Dear reviewer,
Thanks for your careful examination.
We have uploaded a Word file, Please see the attachment.
Best regards,
Jiayu Ou

Reviewer 2 Report
General Comments:
The paper presets an approach to monitor the tool wear condition in the milling cutter. The approach is based on order analysis and stacked sparse auto-encoder. The topic is important in milling industry and the proposed approach is interesting. Clarity of presentation is the main issue that entails a major revision as follows.
Specific Comments:
- Section 2:
- Equation (2) should be referenced.
- The number of necessary layers in the auto encoder should be discussed.
- The size m of vector samples in Line 121 should be discussed.
- It is unclear whether the output in Figure 1 is single or multiple.
- The features and the classification process are not clear.
- Section 3:
- In Step 6, the choice of the optimization algorithm should be justified.
- It is unclear whether the problem would lead to a global minimum.
- Section 4:
- The comparison in Figure 10 is not clear.
- There is confusion in Figure 12 between the number of hidden layers and the number of neurons.
Language Usage:
Language revision is necessary. For example, on Page 2, Line 47, “which cause the signals information cannot fully extract” should be corrected; on Page 4, Line 125, “an activation functions” should be “an activation function”; Line 130, “victor” should be “vector”; Line 131, “which can express as” should be “which can be expressed as”; Line 144, “could express as” should be “can be expressed as”.
Author Response

(The authors gave the same response as above.)

Round 2
Reviewer 1 Report
I am pleased to mention that Authors gave detailed answers to my comments. I suggest Editors to accept the manuscript for publication.
Reviewer 2 Report
The Authors have sufficiently addressed most of the Reviewers' comments. The current version of the manuscript is useful and suitable for publication in MDPI Sensors.